



# Discrete Global Grid System-based Flow Routing Datasets in the Amazon and Yukon Basins

Chang Liao[1], Darren Engwirda[2], Matthew G Cooper[1], Mingke Li[3], and Yilin Fang[4]

[1]Atmospheric, Climate, and Earth Sciences, Pacific Northwest National Laboratory, Richland, WA, USA
[2]T-3 Fluid Dynamics and Solid Mechanics Group, Los Alamos National Laboratory, Los Alamos, NM, USA
[3]Research and Development Group, Geosapiens Inc., Quebec City, QC, Canada
[4]Hydrology Group, Pacific Northwest National Laboratory, Richland, WA, USA

**Correspondence:** Chang Liao (chang.liao@pnnl.gov)

**Abstract.**

Discrete Global Grid systems (DGGs) are emerging spatial data structures widely used to organize geospatial datasets across scales. While DGGs have found applications in various scientific disciplines, including atmospheric science and ecology, their integration into physically based hydrologic models and Earth System Models (ESMs) has been hindered by the lack of flow-routing datasets based on DGGs. In response to this gap, this study pioneers the development of new flow routing datasets using Icosahedral Snyder Equal Area (ISEA) DGGs and a novel mesh-independent flow direction model. We present flow routing datasets for two large basins, the tropical Amazon River Basin and the Arctic Yukon River Basin. These datasets demonstrate the potential of DGGs-based flow routing datasets to enhance the performance of hydrologic models and provide observationally-based flow routing inputs for immediate application to the Amazon and Yukon River Basins. The data are available at https://doi.org/10.5281/zenodo.8377765 (Liao, 2023).

## 1 Introduction

Discrete Global Grid systems (DGGs) are emerging spatial data models that use hierarchical tessellations of cells to partition and address the Earth's surface. DGGs have been widely adopted as a standard data fabric to organize geospatial datasets across various granularities (Goodchild, 1994; Kimerling et al., 1999; Sahr, 2015; Matthew B. J. Purss et al., 2016). DGGs, especially the Icosahedral Snyder Equal Area (ISEA) aperture 3 Hexagon (3H), are used in many disciplines, including Geographic Information System (GIS) (Kevin Sahr, 2019), hydrology (Li et al., 2022), atmospheric science (Randall et al., 2002), and ecology (Ellis et al., 2021; Mechenich and Žliobaitė, 2023). However, DGGs have seen limited adoption in physically-based, spatially distributed hydrologic models and Earth System Models (ESMs) (Li et al., 2022), mainly because ready-for-analysis flow routing datasets based on DGGs are unavailable.

Flow routing datasets are essential for spatially distributed hydrologic models, and they typically rely on two data model paradigms. The first one is the rectangular mesh-based grids, also known as rasters (Esri Water Resources Team, 2011; Wu et al., 2012). This method often requires high-quality digital elevation model (DEM) rasters. It is also subject to several other limitations, including the challenge of coupling with other unstructured mesh-based numerical models. The second one is the





vector-based polylines (Lin et al., 2021), which are often produced through the combination of high-resolution raster-based

and remote sensing product-based methods. However, these polylines often contain various artifacts, including disconnected segments, and thus cannot be directly used across different spatial scales (Huang and Frimpong, 2016). One limitation of this method is the lack of communication between the river and its adjacent riparian zones.

We recently pioneered the ability to generate such datasets using unstructured Model for Prediction Across Scales (MPAS) meshes (Engwirda and Liao, 2021; Liao et al., 2023b). Although MPAS meshes have gained traction in the oceanographic and

30 atmospheric modeling communities, DGGs meshes are also widely adopted across the Earth Sciences and GIS communities. To date, there are no available DGGs-based flow routing datasets that include flow direction information. This is because existing DGGs-based hydrology datasets are often derived by resampling from existing raster-based datasets, which does not support vector-based datasets (Chiranjib Chaudhuri et al., 2021). Besides, most traditional flow direction models in various GIS software only support raster datasets. This highlights the need for a method to natively generate flow direction datasets

within the DGGs-based framework.

Compared to structured rectangular meshes, including latitude-longitude geographic coordinate systems (GCS) and projected coordinate systems (PCS), DGGs have several advantages. These include 1) improved numerical performance for surface and subsurface hydrologic models (Liao et al., 2020); 2) better spatial coverage and consistent spatial resolution for the high latitudes; and 3) more flexibility in spatial resolution due to their hierarchical data structure. Specifically, the ISEA3H

DGGs projection stands out for its benefits to hydrologic models and ESMs. As an equal-area icosahedral DGGs projection, ISEA streamlines calculations of conserved quantities, eliminating the need for post-hoc equal-area reprojection. In addition, the hexagonal grid geometry resolves ambiguity among cell neighborhoods by ensuring uniform adjacency, thereby offering significant advantages in the domain of hydrology.

This study breaks new ground by developing new flow routing datasets using the ISEA3H DGGs and our newly devel-

45 oped mesh-independent flow direction model. We present flow routing datasets for the Amazon and Yukon Basins, which are among the world's largest river basins in the tropics and Arctic, respectively, and play important roles in local, regional, and global climate and ecosystems. These datasets demonstrate the potential of DGGs-based flow routing datasets to enhance the performance of hydrologic models.

## 2 Method

A list of datasets and models used in our workflow to produce DGGs-based flow routing datasets is depicted in Figure 1.





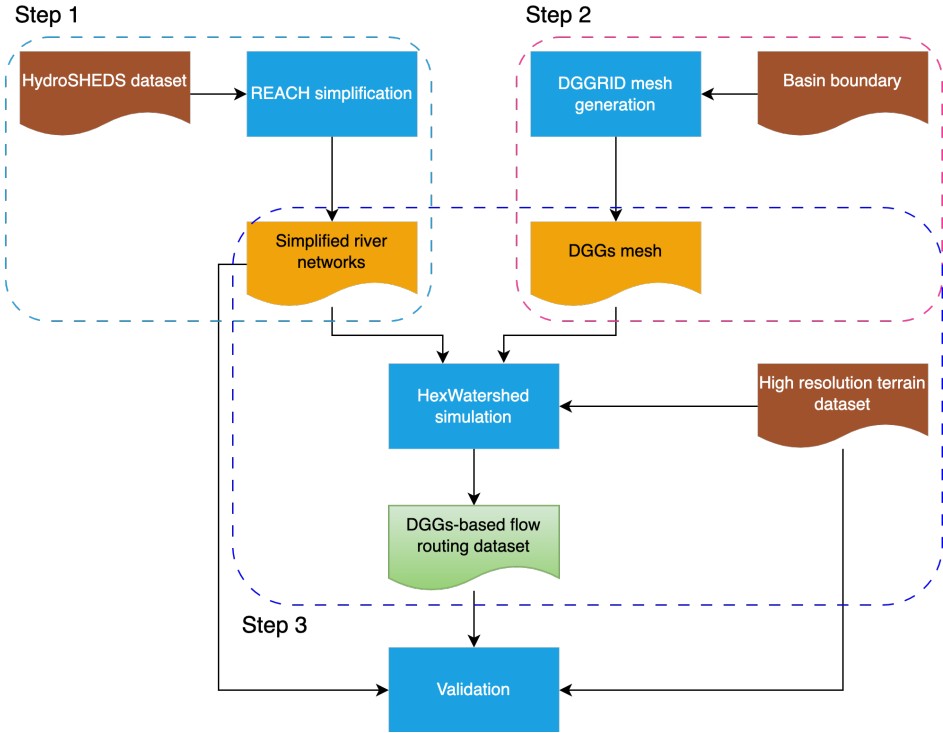

**Figure 1.** Workflow diagram demonstrating the DGGs-based flow routing dataset generation process using three steps (dashed boxes). Step 1: river network simplification using REACH; Step 2: DGGs mesh generation using DGGRID; Step 3: Flow routing modeling using HexWatershed. Brown boxes are user-provided datasets. Orange boxes are intermediate results. The light green box is the final data product.

We first introduce the input datasets used in each step and then the models used. Additional information is provided in the Supplementary Information.

## 2.1 Input datasets

### 2.1.1 Vector river networks

The vector river network datasets of the Amazon and Yukon Basins were obtained from the HydroSHEDS database (Lehner et al., 2008). HydroSHEDS v1 dataset is derived primarily based on elevation data obtained in 2000 by the United States National Aeronautics and Space Administration's (NASA) Shuttle Radar Topography Mission (SRTM). Specifically, we obtained the HydroRiver datasets of South America and the Arctic (https://www.hydrosheds.org/products/hydrorivers). While HydroSHEDS products, including the river networks, may not provide the highest level of accuracy in depicting river and

basin maps, they are widely acknowledged and evaluated for applications at regional and global scales. This dataset is used in Step 1.



### 2.1.2 Vector watershed boundary

The vector Amazon Basin boundary was obtained through NASA's Oak Ridge National Laboratory (ORNL) Distributed Active Archive Center (DAAC) (Emilio Mayorga et al., 2012). The vector Yukon Basin boundary was obtained through HydroBASINS, which is part of the HydroSHEDS product. These datasets are used in Step 2.

### 2.1.3 Raster terrain datasets

High spatial resolution DEM datasets of the Amazon Basin at 30-arc-second ($\sim 1\,km$) were obtained through NASA's ORNL DAAC (Saatchi, 2013). Similar to HydroSHEDS, this DEM was produced as a subset of the SRTM DEM. The flow accumulation and length datasets at the same spatial resolution are used for data validation. Similarly, void-filled DEMs and flow accumulation datasets of the Yukon Basin at 15-arc-second ($\sim 500\,m$) resolution were obtained from the HydroSHEDS. These datasets are used in Step 3.

## 2.2 Models

Our workflow primarily leverages three software models to produce the DGGs-based flow routing datasets. The models are run in sequence in three steps: 1) the REACH model pre-processes the vector river networks, i.e., HydroSHEDS, to produce the simplified river networks; 2) the DGGRID model generates the DGGs mesh using the basin boundary; and 3) the HexWatershed model generates the flow routing datasets using outputs from Step 1 and 2. Descriptions of each model and step are provided below.

### 2.2.1 HydroSHEDS river network simplification using REACH

Because the full HydroSHEDS river network dataset contains millions of river channels that range between a few to thousands of kilometers, they cannot be represented equally in hydrologic and Earth system models. For example, any river channel less than $10\,km$ in length cannot be represented well if the mesh cell resolution is also $10\,km$. To address this challenge, we used the REACH library (Engwirda, 2023) to pre-process (simplify) the HydroSHEDS river network. In this step, only major river channels and tributaries resolvable at scales of interest are preserved. REACH employs a greedy network simplification algorithm in which the maximal set of river reaches is processed in priority order of increasing upstream catchment area. River reaches are removed incrementally if they meet the following criteria: 1) their length is shorter than a user-defined tolerance, or 2) they are geometrically closer to another, higher priority reach segment than a user-defined tolerance. Upon removal of a given river reach, the downstream network is simplified — merging any newly contiguous segments into 'super-reaches' and updating their associated priorities. While heuristic in nature, this greedy approach leads to simplified river vector networks that are appropriate for both hydrological analysis and unstructured mesh generation, with the network pruned in a least-catchment-area-first manner. This retains hydrologically important reaches while removing geometrical features smaller than the desired mesh scale to ensure compatibility between the flow network and the computational grid.

In practice, the user-defined tolerance is often set as the mesh cell's spatial resolution, which can vary in space as well. In this study, because we use four resolution levels from the DGGRID model, the corresponding resolutions are used as the user-defined tolerance parameters. Figure 2 illustrates the simplified HydroSHEDS river networks in the Amazon Basin.

Simplified flowline by REACH

**Figure 2.** Simplified river networks using the REACH library at DGGRID ISEA3H level 10 to 13 resolutions in the Amazon Basin. As these datasets are extracted from the global HydroSHEDS river networks, there may be isolated river segments near the boundary of the basin.





### 2.2.2 Mesh generation using DGGRID

DGGRID is an open-source library developed by Kevin Sahr in 2003, mainly used for generating and manipulating DGGs with diverse configurations (Sahr, 2015). The DGGRID library provides various grid geometry options, including triangles, diamonds, and hexagons. It also allows for specifying multiple refinement ratios between successive resolutions, customized orientation relative to the Earth's surface, and different projection methods when generating the grids such as the ISEA and FULLER projections (Sahr, 2015). A list of parameters to define the mesh generation process is summarized in (Table A1). For the complete list of parameters, please refer to the DGGRID user manual (Sahr, 2015).

The DGGRID version 7.0 was used in our study to generate the ISEA Aperture 3 Hexagonal (ISEA3H) meshes with the default orientation. A total of five resolution levels from 10 to 14 are defined (Table 1). The level 14 mesh was used for validation only.

| Resolution level | Internode spacing (km) | Mean resolution ($\sqrt{\text{area}}$, km) |
| --- | --- | --- |
| 10 | 31.7596 | 29.42 |
| 11 | 18.341 | 16.99 |
| 12 | 10.5871 | 9.81 |
| 13 | 6.11367 | 5.66 |
| 14 | 3.52911 | 3.26 |

**Table 1.** The DGGRID mesh generation resolutions used to produce the flow routing datasets for Amazon and Yukon.

The four spatial resolutions (levels 10-13) were selected because most large-scale hydrologic models and Earth System Models run at approximately 0.5-degree ($\sim 50$ km at the equator) spatial resolution, which is similar to resolution level 10, while many large-scale hydrologic models run at spatial resolutions of $\sim 5$ km, similar to resolution level 13. These four spatial resolutions, therefore, cover a wide range of hydrologic model applications.

Once the DGGRID v7.0 model is built from its C/C++ source code, it can be used as a library to be directly called by the HexWatershed model through several Application Programming Interface (API) (Liao, 2022a). As a result, Step 2 can be run as part of Step 3.

### 2.2.3 Flow direction modeling using HexWatershed

HexWatershed is a mesh-independent flow direction model for hydrologic models. Unlike most flow direction models that only support structured rectangle meshes, HexWatershed supports both structured and unstructured meshes. HexWatershed includes the state-of-science topological relationship-based river network representation and depression removal methods to generate high-quality flow routing datasets across scales (Liao and Cooper, 2023; Liao, 2022a). These methods allow the embedding of river networks and other hydrologic features within the flow routing map from regional to global scales. To achieve this, HexWatershed uses a two-step approach to model flow direction. First, it uses the mesh-river network intersection to build the topological relationship between mesh cells and river channels (e.g., upstream-downstream channel cells). Next, it uses a



hybrid stream burning-depression filling algorithm to generate the flow direction between all the mesh cells. This step will first define the elevation and flow direction of the river channels and then process the remaining mesh cells. Additional explanations of these techniques are provided in the Supplementary Information and can be found in our two-part series of studies (Liao et al., 2023a, b).

The computational geometry algorithms within HexWatershed accept all types of mesh cells (e,g., rectangle, hexagon, tri-
125 angle, etc.), and the depression removal algorithms automatically consider different numbers of neighbors when defining flow directions. Therefore, HexWatershed is mesh-independent and supports both structured and unstructured meshes.

In this study, we extended HexWatershed to support the DGGRID mesh type. Specifically, we implemented several APIs to set up a DGGRID model run and convert the DGGRID outputs to the HexWatershed model data structure (Step 2). Then we run HexWatershed v3.0 to generate flow routing datasets using the DGGRID-generated ISEA3H meshes at four different
spatial resolutions (Table 1). For each spatial resolution, the HexWatershed model simulation includes the following steps:

    a Prepare all the input datasets (outputs from Step 1) and binaries (DGGRID and HexWatershed C++ binaries) into a workspace folder;

    b Call the PyFlowline Python package (Liao et al., 2023a; Liao and Cooper, 2023) to generate the conceptual river networks. PyFlowline is a core component in the HexWatershed model. This step includes three sub-steps:

– Pre-process the vector river network datasets, i.e., simplified HydroSHEDS river networks from Step 1. This step further processes the river networks, including re-building the stream segment indices and (Strahler) orders;

        – Generate the DGGRID configuration file and run the DGGRID model to generate the DGGs mesh file. This is also the Step 2 in Figure 1;

        – Model the conceptual river networks using the topological relationship-based reconstruction method.

c Assign elevation to the mesh cells based on raster DEM and each mesh cell boundary (Liao et al., 2022). A zonal mean resampling method is used by default;

    d Conduct the depression removal. This step includes two sub-steps (Liao et al., 2023b):

        – Run the topological relationship-based stream burning on the river cells and their riparian zone cells using outputs from Step 3b and 3c;

– Run the revised priority-flood depression filling for the remaining mesh cells.

    e Export and visualize the model outputs, including the flow direction map and other flow routing parameters (Liao, 2022b).

Last, spatial visualizations were produced using Python packages, including Geospatial Data Abstraction Library (GDAL) (GDAL/OGR contributors, 2019) and PyEarth (Liao, 2022b).



## 3  Data record

These datasets contain four collections of flow-routing datasets corresponding to four spatial resolutions for both the Amazon and Yukon Basins (Liao, 2023). Within each collection, several files are provided, with a README file explaining each file. The results from resolution level 10 are used here for illustration purposes. The Supplementary Information provides visualizations of all four dataset collections with zoom-in views. All the spatial datasets are provided using the GeoJSON file format with the GCS spatial reference.

### 3.1  Surface elevation

The **variable_polygon.geojson** file is a polygon-based GeoJSON data file. The attribute "elevation" stores the modeled zonal mean surface elevation for each DGGRID mesh cell after the depression removal (Figures 3 and 4).

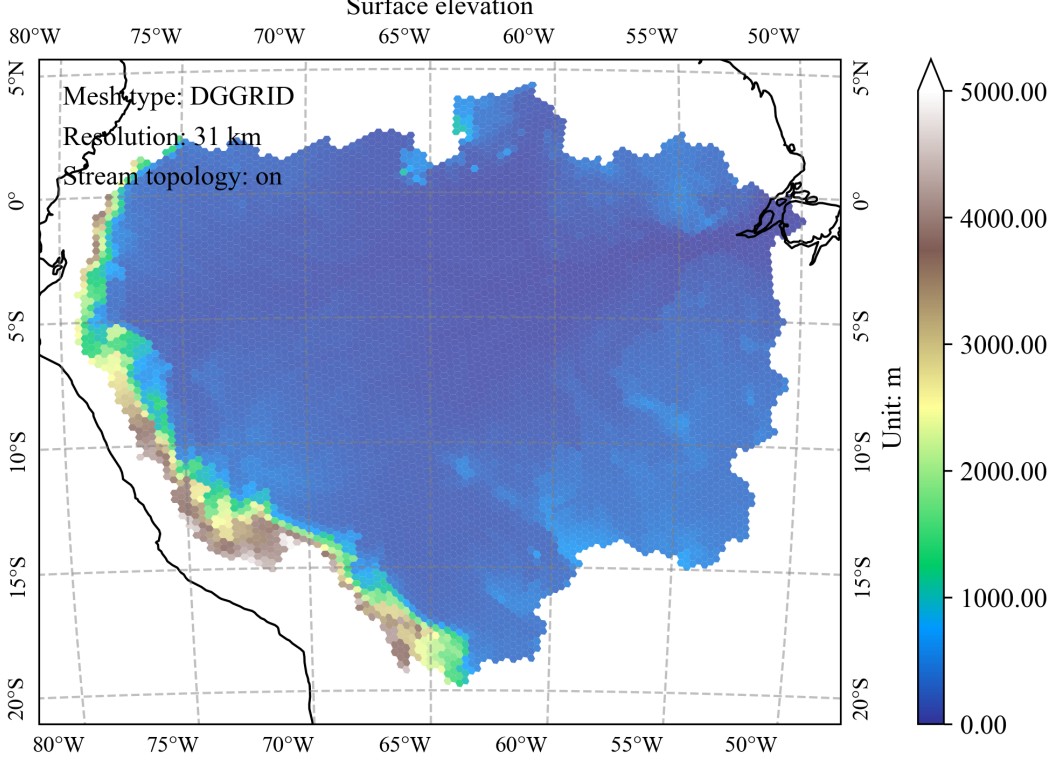

**Figure 3.** Spatial distribution of modeled surface elevation at DGGRID ISEA3H level 10 resolution in the Amazon Basin (unit: m).



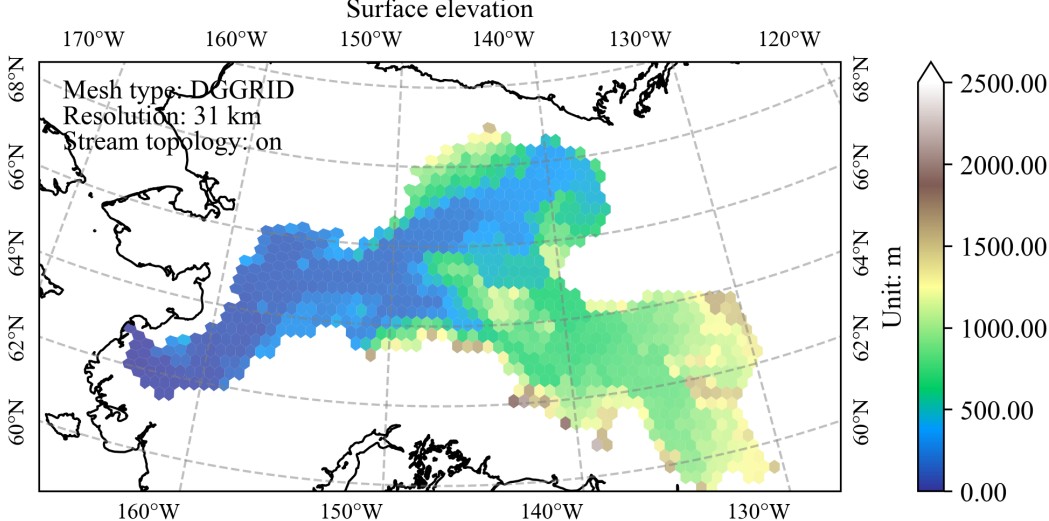

**Figure 4.** Spatial distribution of modeled surface elevation at DGGRID ISEA3H level 10 resolution in the Yukon Basin (unit: m).

## 3.2 Surface slope

The **variable_polygon.geojson** file is a polygon-based GeoJSON data file. The attribute "slope" stores the modeled between-cell surface slope based on the depression-free elevation difference and cell center-to-cell center distance (Figures 5 and 6).

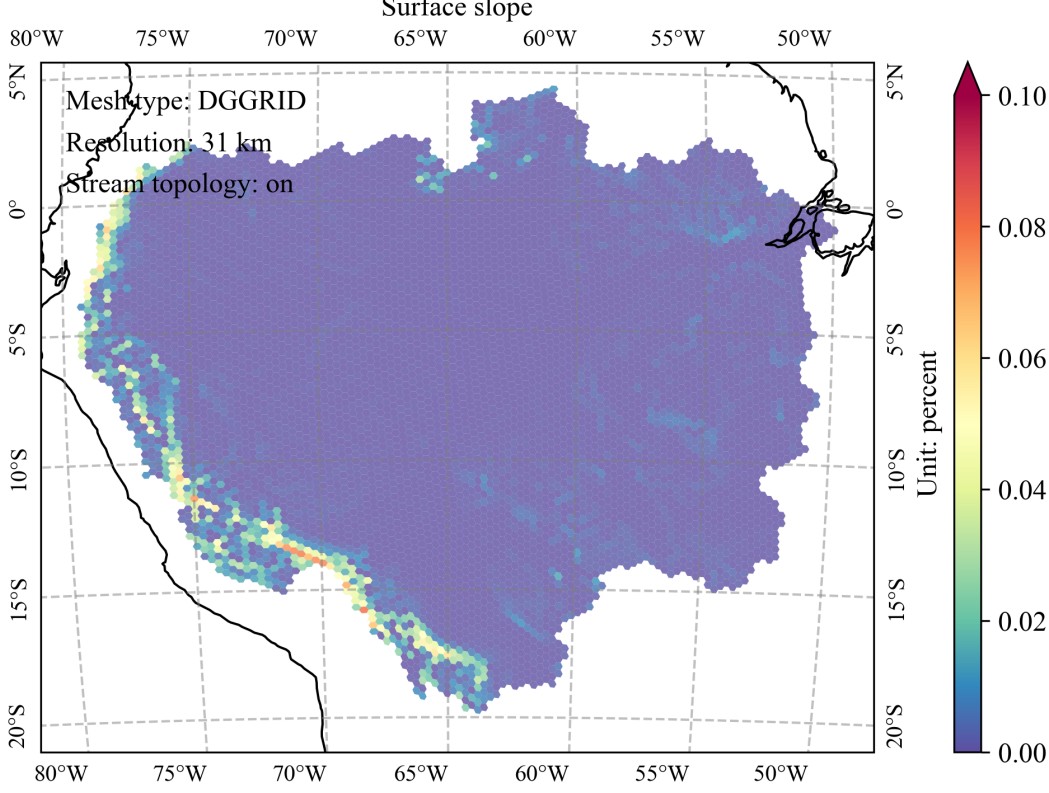

**Figure 5.** Spatial distribution of modeled mesh cell center to cell center slope at DGGRID ISEA3H level 10 resolution in the Amazon Basin (unit: percent).





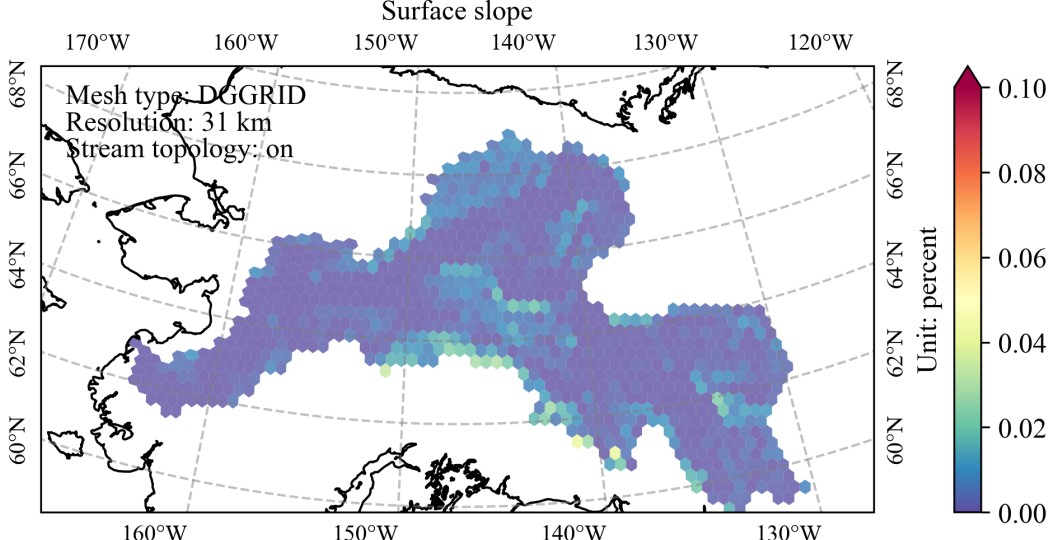

**Figure 6.** Spatial distribution of modeled mesh cell center to cell center slope at DGGRID ISEA3H level 10 resolution in the Yukon Basin (unit: percent).

## 3.3 Flow direction

The **flow_direction.geojson** is a polyline-based GeoJSON data file. Each polyline feature defines the single flow direction (the steepest slope) from one DGGRID mesh cell center to its downslope/downstream mesh cell center (Figures 7 and 8).

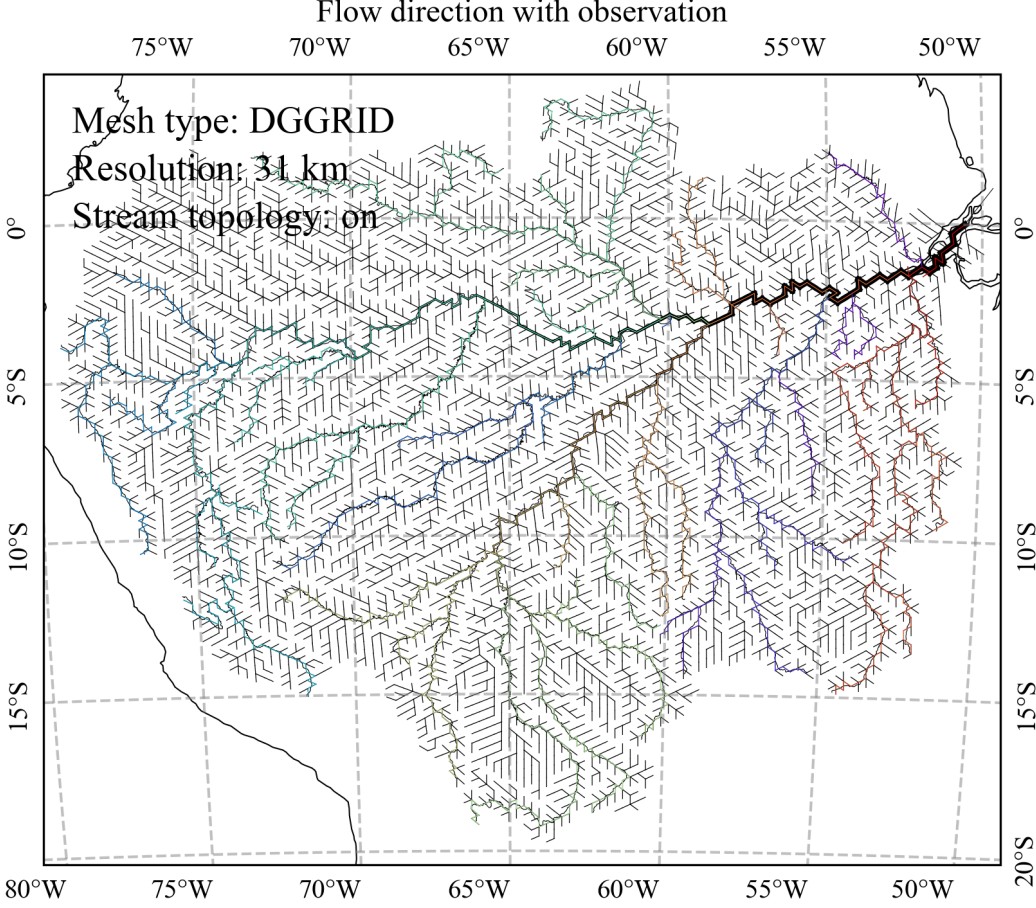

**Figure 7.** Modeled flow direction at DGGRID ISEA3H level 10 resolution in the Amazon Basin. Black straight lines are cell-to-cell conceptual flow direction. Line thickness is scaled with drainage area. Colored and curved black lines are conceptual and simplified HydroSHEDS river networks.





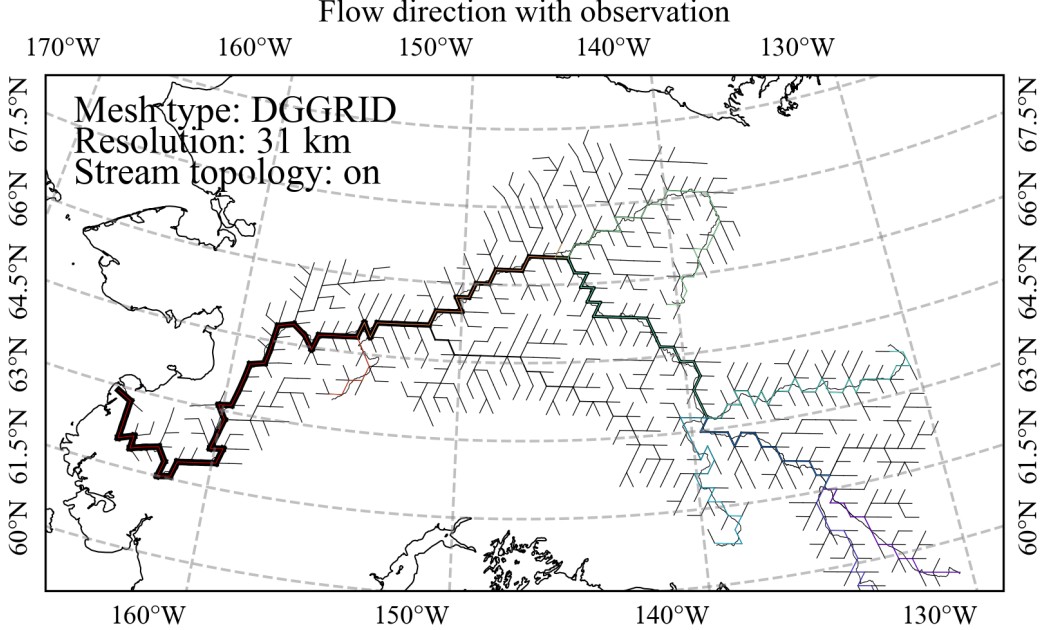

**Figure 8.** Modeled flow direction at DGGRID ISEA3H level 10 resolution in the Yukon Basin. Black straight lines are cell-to-cell conceptual flow direction. Line thickness is scaled with drainage area. Colored and curved black lines are conceptual and simplified HydroSHEDS river networks.

### 3.4 Drainage area

The **variable_polygon.geojson** is a polygon-based GeoJSON data file. The attribute "drainage" stores the modeled total upstream drainage area of each mesh cell (including its own area) (Figures 9 and 10).

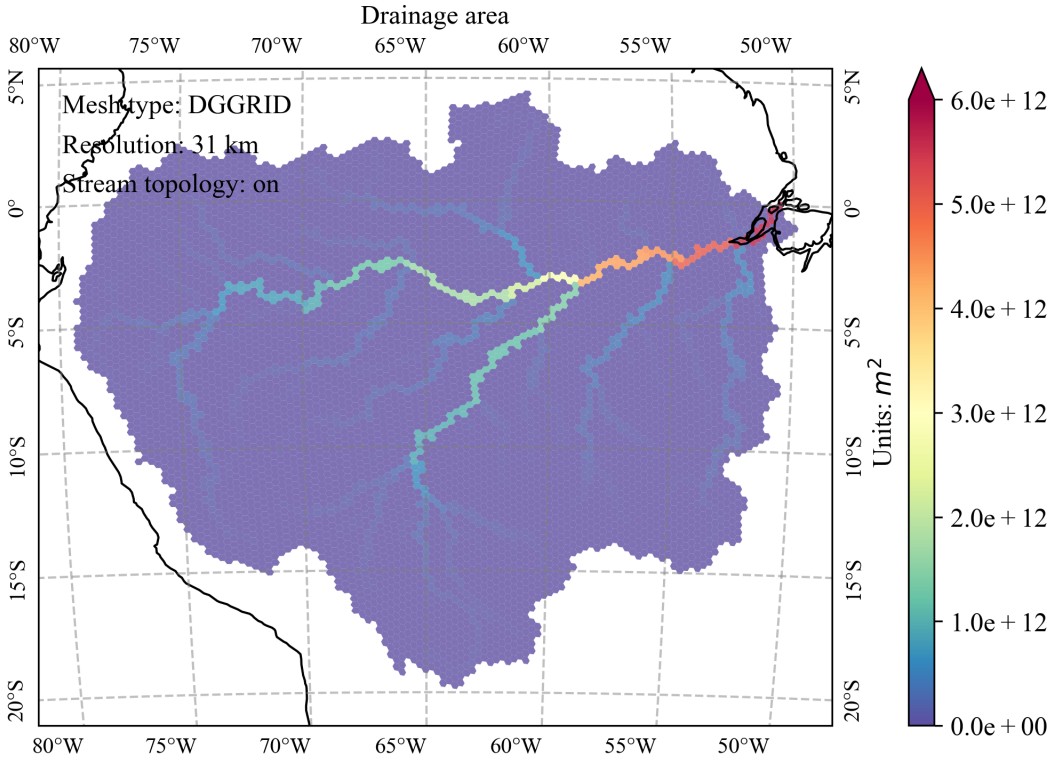

**Figure 9.** Modeled drainage area at DGGRID ISEA3H level 10 resolution in the Amazon Basin (units: m$^2$).

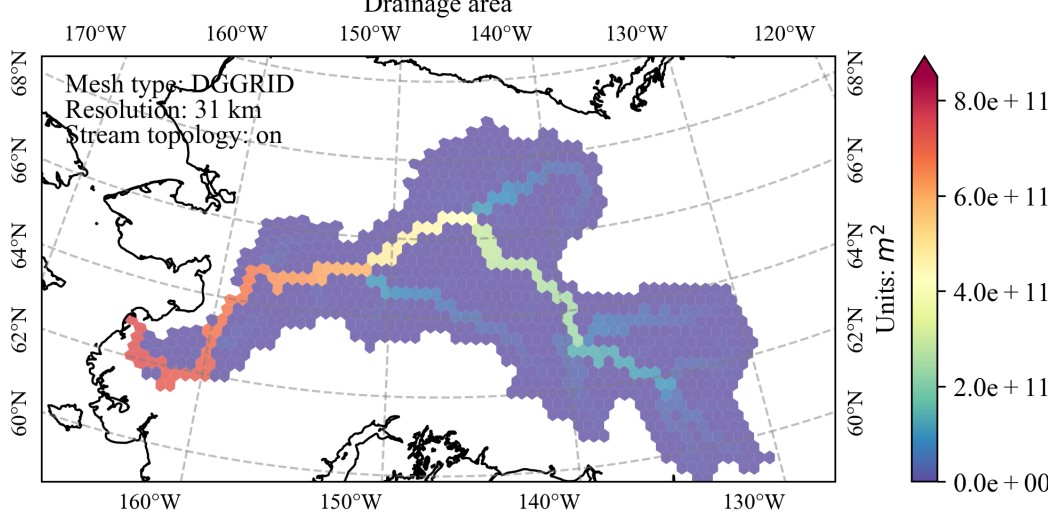

**Figure 10.** Modeled drainage area at DGGRID ISEA3H level 10 resolution in the Yukon Basin (units: m$^2$).



## 3.5 Travel distance

The **variable_polygon.geojson** is a polygon-based GeoJSON data file. The attribute "travel_distance" stores the modeled
travel distance from each mesh cell to the basin outlets (Figures 11 and 12). This term is also often referred to as downstream
flow length.

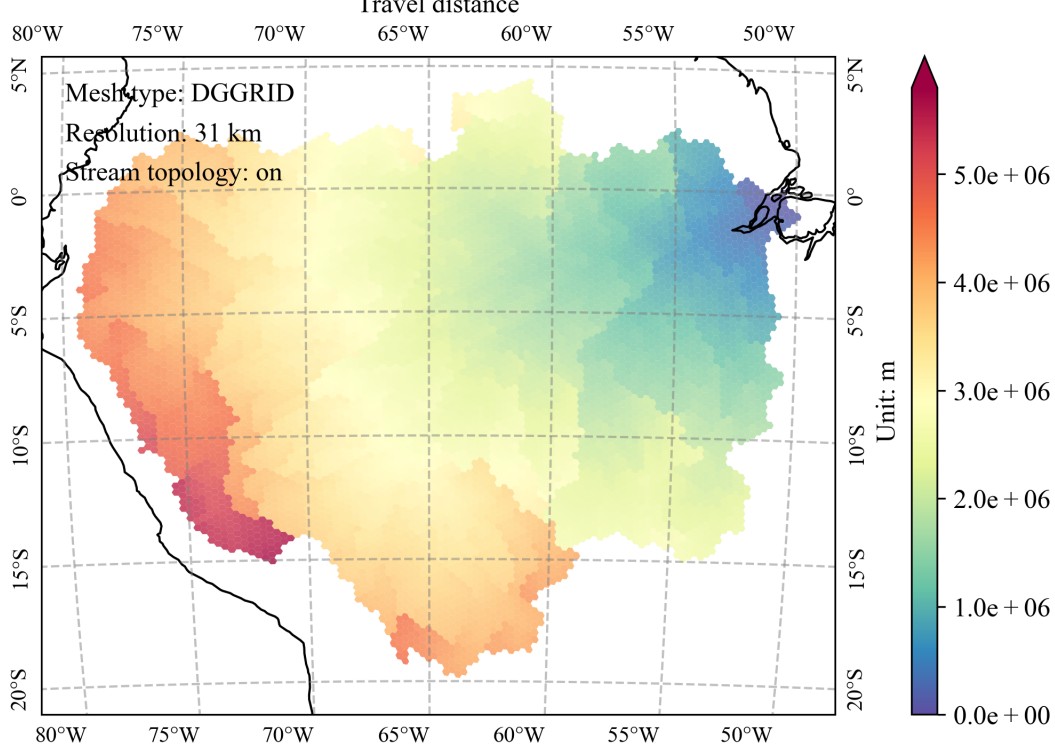

**Figure 11.** Modeled travel distance to the basin outlet at DGGRID ISEA3H level 10 resolution in the Amazon Basin (unit: m).



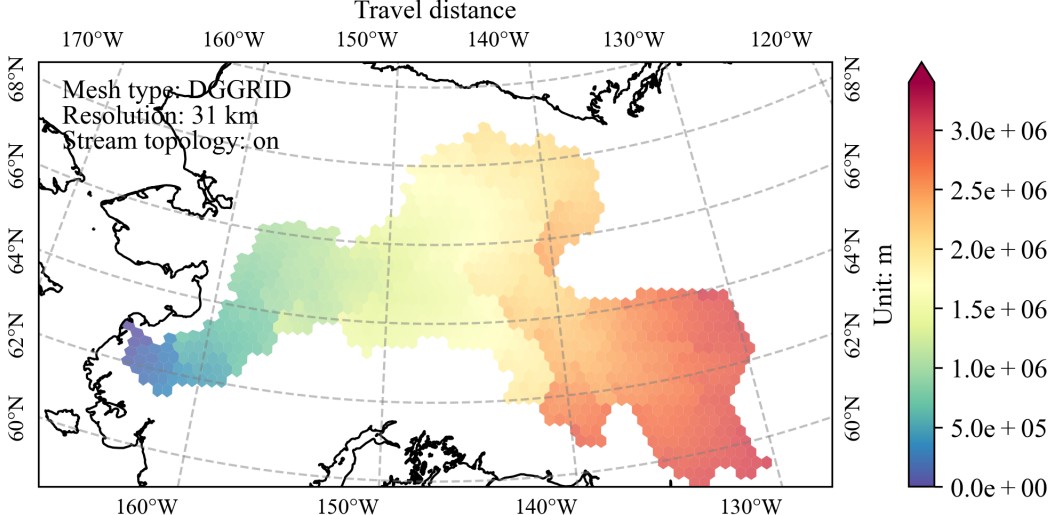

**Figure 12.** Modeled travel distance to the basin outlet at DGGRID ISEA3H level 10 resolution in the Yukon Basin (unit: m).

## 4 Technical Validation

Because the fundamental mesh structures differ from most existing datasets, we mainly rely on spatial patterns and geostatistics to evaluate our datasets. Different strategies are used for different data records. We primarily evaluate our datasets using existing

flow routing datasets, i.e., HydroSHEDS products, the LBA-ECO CD-06 Amazon River Basin Land and Stream Drainage Direction and DEM datasets (Emilio Mayorga et al., 2012; Saatchi, 2013). Because our datasets include four different spatial resolutions, special attention was paid to the consistency across different spatial resolutions.

### 4.1 Surface elevation

We employed a sphere resampling method to assess the surface elevation data through the following steps: (1) Utilizing the

DGGRID ISEA3H level 14 mesh (the highest resolution in the current workflow) as the sampling pool; (2) Randomly selecting N cells as points of interest and recording their center locations; (3) Extracting elevation values from the data records and existing DEM datasets based on the chosen N longitude/latitude pairs. A scatterplot featuring N = 500 sampling points demonstrates that the modeled elevations closely match those of the existing high-resolution raster DEMs.



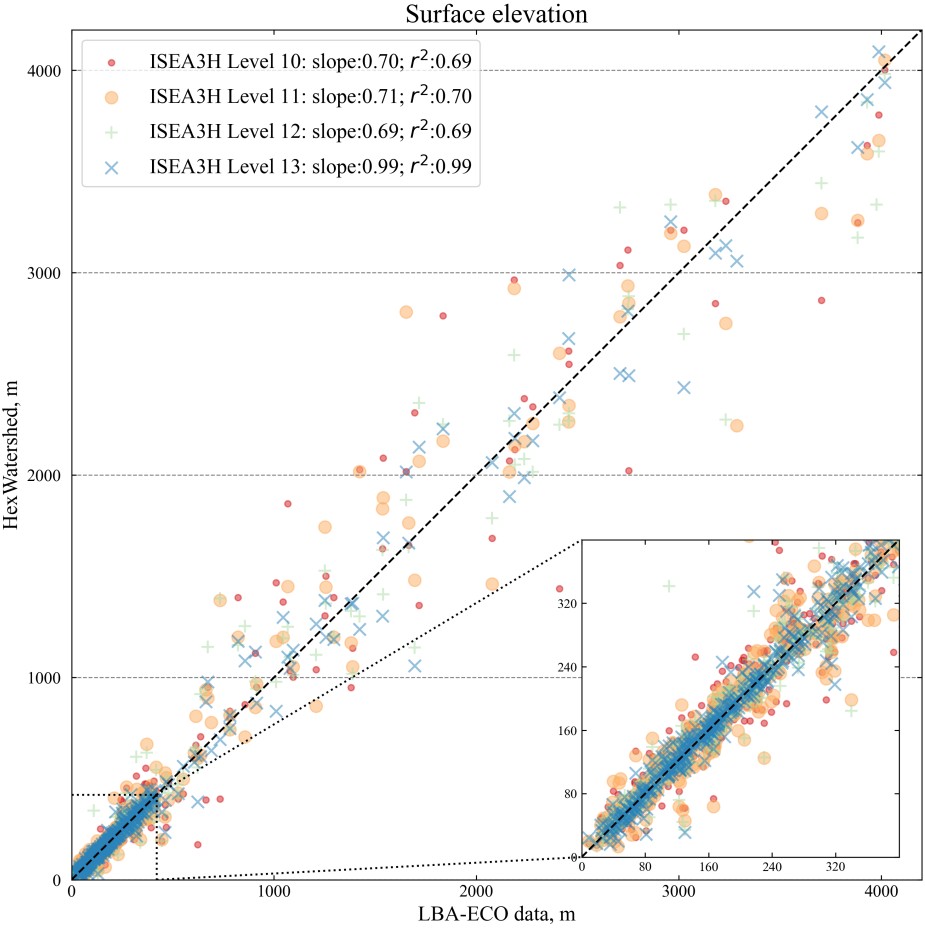

**Figure 13.** Validation of modeled surface elevation in the Amazon Basin from four DGGRID mesh resolutions. The x-axis is the sampled elevation from the LBA-ECO DEM datesets. The y-axis is the sampled surface elevation from our records (unit: m). The mini-plot is a zoom-in view of the lower left.

The modeled surface elevation in the Yukon Basin is slightly worse than that in the Amazon Basin. One reason is that the spatial resolution of LBA-ECO DEM (30-second) is twice that of the HydroSHEDS DEM (15-second). Meanwhile, the Amazon Basin has relatively flat terrain compared with the Yukon Basin. This leads to different biases during the zonal mean resampling procedure.



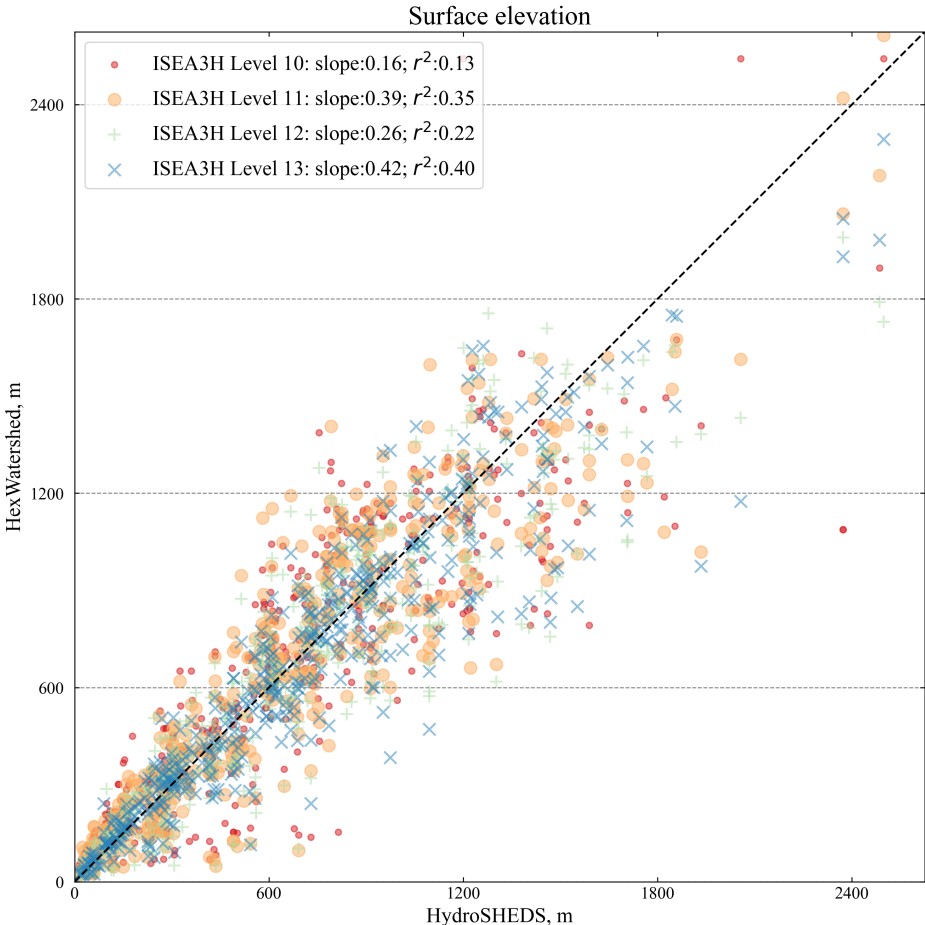

**Figure 14.** Validation of modeled surface elevation in the Yukon Basin from four DGGRID mesh resolutions. The x-axis is the sampled elevation from the HydroSHEDS DEM datesets. The y-axis is the sampled surface elevation from our records (unit: m).

## 4.2 Flow direction

Given that flow direction is a vector field, a direct comparison between the modeled flow directions and existing D4/D8-based
flow direction datasets is not feasible. Instead, we conducted a visual examination of the modeled flow directions using the simplified HydroSHEDS river networks. As depicted in Figures 7 and 8, the modeled flow directions consistently align with the simplified HydroSHEDS river networks across all four resolutions, consistent with our previous study (Liao et al., 2023b). Additionally, flow direction can be indirectly validated using the drainage area since they are closely interconnected.





## 4.3 Drainage area

The sphere resampling method (Section 4.1) could not be directly applied to the drainage area due to issues related to resolution mismatch and spatial dependence. As an alternative, we conducted a comparative analysis using major tributaries along the Amazon River and Yukon River, including their mouths.

In the Amazon Basin, we selected seven tributary outlets, and their locations are provided in the Supplementary Information section. The scatterplot shows that the modeled drainage areas are consistent with the existing LBA-ECO drainage datasets 200 (Figure 15).

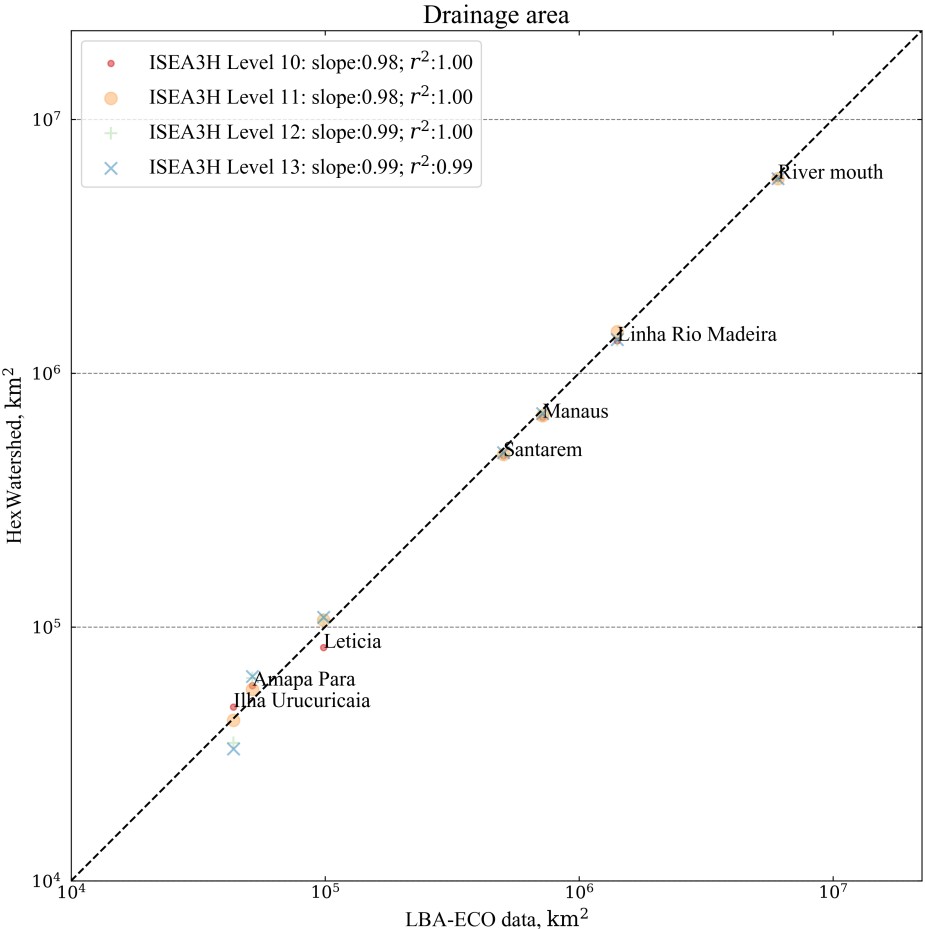

**Figure 15.** Validation of modeled drainage area of seven tributaries (including the river mouth) along the Amazon River from four DGGRID mesh resolutions. The x-axis is the drainage area from the LBA-ECO CD-06 Amazon River Basin Land and Stream Drainage Direction datesets (converted from flow accumulation). The y-axis is the modeled drainage area (units: $km^2$). Both the x and y axes are in the log scale.



In the Yukon Basin, we selected six tributary outlets, and their locations are provided in the Supplementary Information section. Among these tributaries, only the modeled drainage area at resolution level 10 at the Kooyukuk River is underestimated (Figure 16).

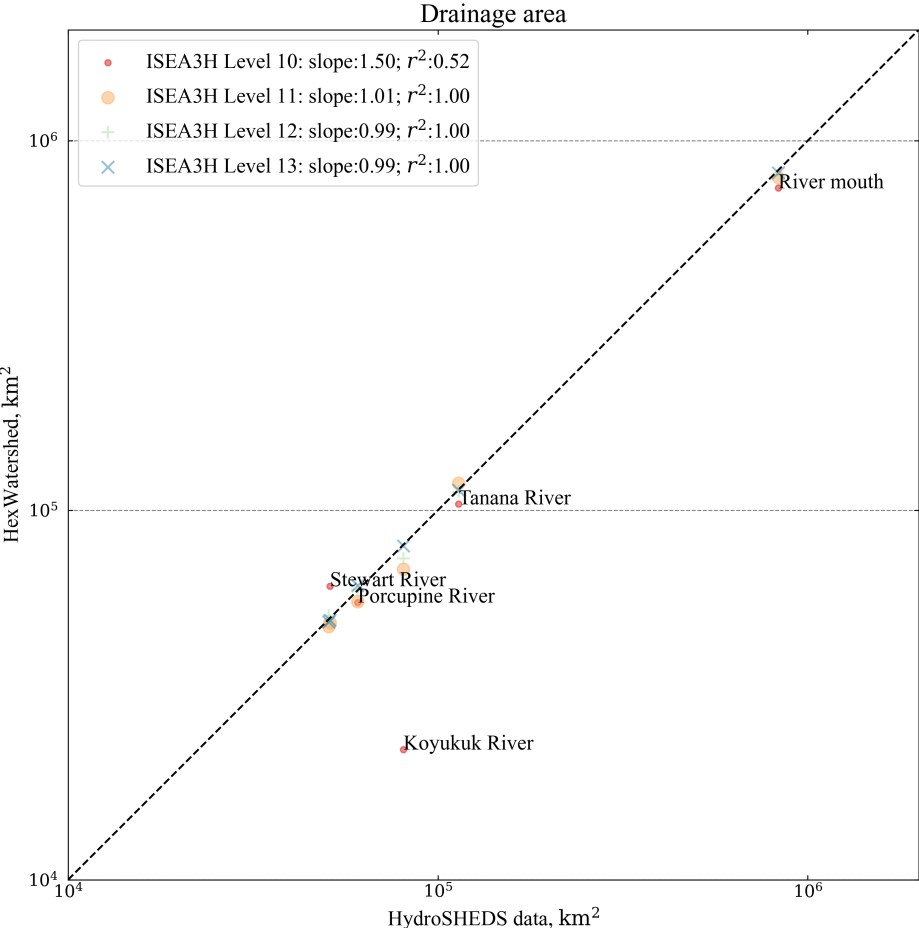

**Figure 16.** Validation of modeled drainage area of six tributaries along the Yukon River from four DGGRID mesh resolutions. The x-axis is the drainage area from the HydroSHEDS datesets. The y-axis is the modeled drainage area (units: km$^2$). Both the x and y axes are in the log scale.

## 4.4 Travel distance

Similar to the drainage area, we evaluated the modeled travel distance using the selected tributary outlets, excluding the river mouths. In the Amazon Basin, the scatterplot shows that the modeled travel distances are consistent with the existing LBA-ECO CD-06 flow length datasets (Figure 17). However, the modeled travel distances are slightly higher than the LBA-ECO datasets. This is possibly caused by the additional length added near the Amazon River delta region.



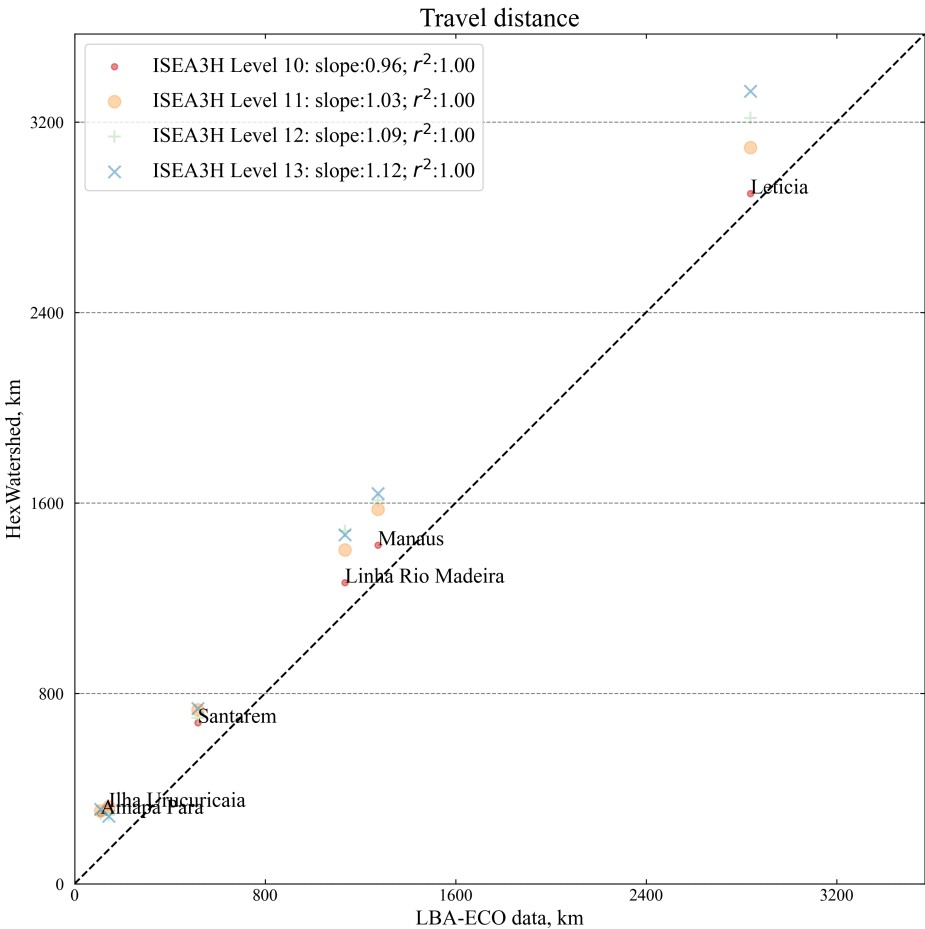

**Figure 17.** Validation of modeled travel distance of six tributaries along the Amazon River from four DGGRID mesh resolutions. The x-axis is the travel distance from the LBA-ECO CD-06 Amazon River Basin Land and Stream Drainage Direction datesets. The y-axis is the modeled travel distance (unit: km).




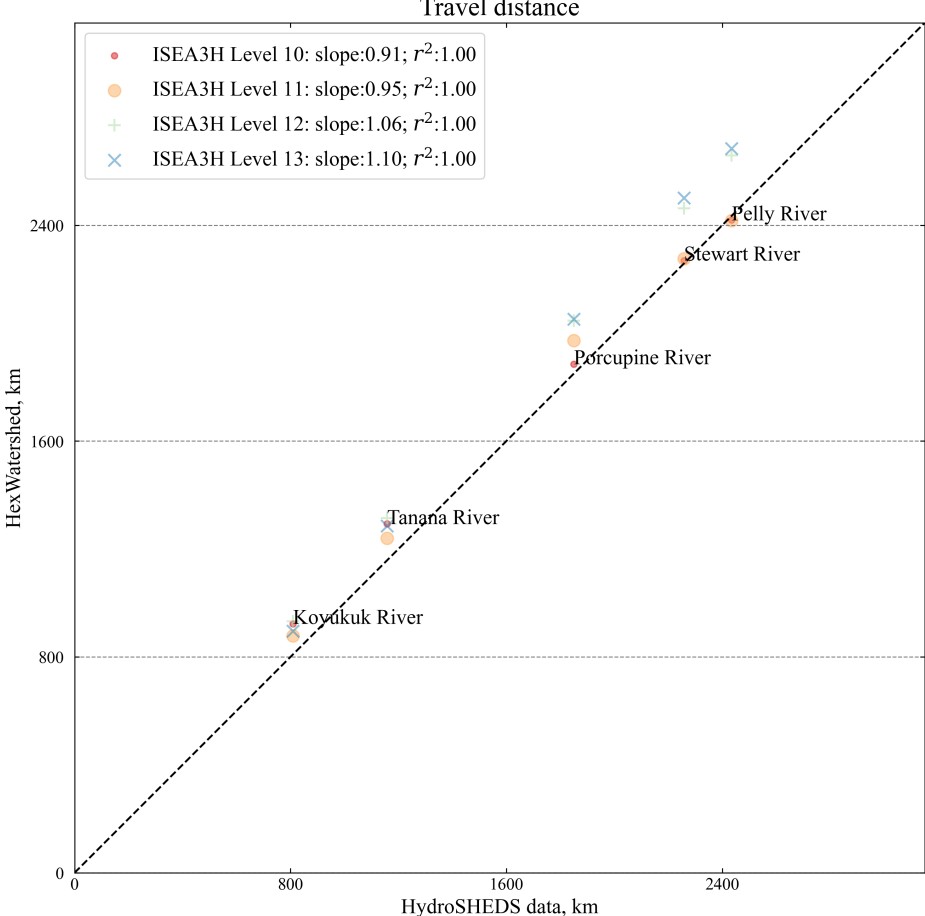

**Figure 18.** Validation of modeled travel distance of six tributaries along the Yukon River from four DGGRID mesh resolutions. The x-axis is the travel distance from the HydroSHEDS datasets. The y-axis is the modeled travel distance (unit: km).

## 5 Discussion

### 5.1 Limitation

Although our datasets represent an important step forward in flow routing capability, they are not without limitations. Through our practice, we have identified the following limitations and provided corresponding solutions for future improvements:

1. As a pioneering dataset, our dataset records were not produced from a unified data source, especially for the DEM component. This is because nearly all the existing global DEM datasets use the GCS spatial reference, and the quality of DEM gradually decreases from the equator to the high latitudes due to the spatial distortion (Liao et al., 2020). This is



also part of the reason that the modeled surface elevation in the Yukon Basin is not as good as that in the Amazon Basin. However, the workflow we developed is generally robust even though there are potential issues in the DEM datasets.

2. The HexWatershed model does not accommodate braided rivers currently, potentially introducing uncertainty in the flow direction, particularly in regions like the Amazon River mouth and delta. However, since most hydrologic models also lack support for braided rivers, this limitation is not considered critical at this time.

3. The modeled drainage area and travel distance were slightly higher than observed in the Amazon Basin (Figure 17). We interpret this as due to the inclusion of the complex delta region, which suggests a similar bias could arise in similar deltaic regions worldwide. To address this, it is recommended to use high-resolution meshes (such as the ISEA3H resolution level 13) to mitigate its impact on model performance.

4. Our method does not consider large lake waterbody in the workflow. Therefore, these datasets may not be suitable for hydrologic applications that focus on lake routing. We plan to explicitly consider lakes, especially large lakes, in future developments.

5. Due to computational constraints and input dataset quality, we only generated flow routing datasets at four spatial resolutions in the Amazon and Yukon Basins. To evaluate model performance and suitability at finer spatial resolutions (e.g., < 5 km), additional simulations are needed. Additionally, a global-scale dataset will be made available once computational efficiency has been enhanced.

## 5.2 Usage

The datasets are primarily stored using the JavaScript Object Notation (JSON) and GeoJSON formats. Some datasets are also provided in the (Geo)Parquet file format tailored for high-performance operations and visualizations. Most scientific programming languages, including Python, C++, R, and MATLAB, provide functions or public libraries to read these file formats. The datasets are distributed with global coverage, but users can extract portions of the dataset using GIS operations. For example, users can extract sub-basins for regional hydrologic simulations or convert them to other common scientific file formats, including the Network Common Data Format (NetCDF) or Hierarchical Data Format (HDF).

These datasets are suitable for regional and large-scale spatially distributed hydrologic and river routing models, including the Model for Scale Adaptive River Transport (MOSART) (Li et al., 2013). Additionally, users can derive other flow routing parameters like the Topographic Wetness Index (TWI) and Manning's Roughness Coefficient (n) from these datasets.

## 6 Conclusions

We have produced pioneering ISEA3H DGGs-based hierarchical flow routing datasets in the Amazon Basin and Yukon Basin, available at four spatial resolutions (29.42 km, 16.99 km, 9.81 km, and 5.66 km). Extensive evaluation confirms their consistency with existing high-resolution terrain data and HydroSHEDS river networks. Because our method is mesh-independent,



similar flow routing datasets can also be generated on DGGs with various configurations or even other unstructured grid meshes. Adoption of these datasets by hydrologic models will enhance the performance of spatially distributed hydrological models of these two basins and similar regions worldwide.

## 7 Code availability

The REACH tool can be accessed from the GitHub repository: https://github.com/dengwirda/reach. The DGGRID model can be accessed from the GitHub repository: https://github.com/sahrk/DGGRID. The HexWatershed model can be installed through the Conda Python platform: https://anaconda.org/conda-forge/hexwatershed (Liao, 2022a; Liao and Cooper, 2022). The source code to reproduce the datasets and figures is stored in the GitHub repository: https://github.com/changliao1025/liao_2023_scidata_dggs.

## 8 Data availability

The datasets are stored in the Zenodo repository: https://zenodo.org/record/8377765 (Liao, 2023).

## Appendix A: Model configurations

### A1 DGGRID model configurations

### A2 HexWatershed model configurations

| Variable name | Data type | Data format | Note |
|---|---|---|---|
| area | float | GEOJSON | Geodesic area |
| elevation | float | GEOJSON | Mean elevation after the depression removal |
| slope | float | GEOJSON | Slope between mesh cell in the flow direction |
| flow direction | Not applicable | JSON/GEOJSON | Dominant flow direction with the steepest slope |
| drainage area | float | GEOJSON | Geodesic area-based |
| travel distance | float | GEOJSON | Cell center to cell center distance-based |

**Table A2.** List of data records produced by HexWatershed in the Amazon Basin.

## Appendix B: HexWatershed method description

### B1 PyFlowline

The PyFlowline model is a core submodule within the HexWatershed model. PyFlowline generates the mesh cell-based conceptual river networks using three steps: (1) flowline simplification, which removes undesired flowlines and builds the topolog-



ical relationships; (2) mesh generation, which creates customized meshes based on model configuration. For example, it now
supports APIs to generate a DGGRID mesh; (3) topological relationship reconstruction. This algorithm uses the intersection
between flowlines and mesh cells to reconstruct the cell-to-cell topological relationships.

## B2    HexWatershed

HexWatershed is a mesh-independent flow direction model and fully supports all the mesh types generated by PyFlowline.
It defines flow direction using a two-step approach. First, it uses a hybrid breaching-filling stream-burning method to define
the flow direction for river networks and their riparian zones. Second, it uses a revised priority-flood algorithm to conduct
depression filling and defines the flow direction for the remaining mesh cells. A list of other flow routing parameters are
generated through this process.

A video describing the hybrid stream burning and depression filling algorithm is provided using the ISEA3H resolution 11
simulation animation.



**Appendix C: Full data record visualization**

## C1   Amazon Basin

Surface elevation

![Spatial distribution of modeled surface elevation at four DGGRID resolutions in the Amazon Basin]

**Figure C1.** Spatial distribution of modeled surface elevation at DGGRID ISEA3H level 10 to 13 resolutions in the Amazon Basin (unit: m).



**Figure C2.** Spatial distribution of modeled mesh cell center to cell center slope at DGGRID ISEA3H level 10 to 13 resolutions in the Amazon Basin (unit: percent).



Flow direction with observation

**Figure C3.** Modeled flow direction at DGGRID ISEA3H level 10 to 13 resolutions in the Amazon Basin. Black lines are cell-to-cell flow direction. Line thickness is scaled with drainage area. Colored and detailed black lines are conceptual and simplified HydroSHEDS river networks.



Flow direction with observation



**Figure C4.** Zoom-in views of modeled flow direction at DGGRID ISEA3H level 10 to 13 resolutions in the Amazon Basin near Manaus. Black lines are cell-to-cell flow direction. Line thickness is scaled with drainage area. Colored and detailed black lines are conceptual and simplified HydroSHEDS river networks. The base images are Openstreet Map contributors 2024. Distributed under the Open Data Commons Open Database License (ODbL) v1.0.



**Figure C5.** Modeled drainage area at DGGRID ISEA3H level 10 to 13 resolutions in the Amazon Basin (units: m$^2$).



**Figure C6.** Modeled travel distance to the basin outlet at DGGRID ISEA3H level 10 to 13 resolutions in the Amazon Basin (unit: m).





## C2  Yukon Basin

Surface elevation



**Figure C7.** Spatial distribution of modeled surface elevation at DGGRID ISEA3H level 10 to 13 resolutions in the Yukon Basin (unit: m).





Surface slope



**Figure C8.** Spatial distribution of modeled mesh cell center to cell center slope at DGGRID ISEA3H level 10 to 13 resolutions in the Yukon Basin (unit: percent).

Flow direction with observation

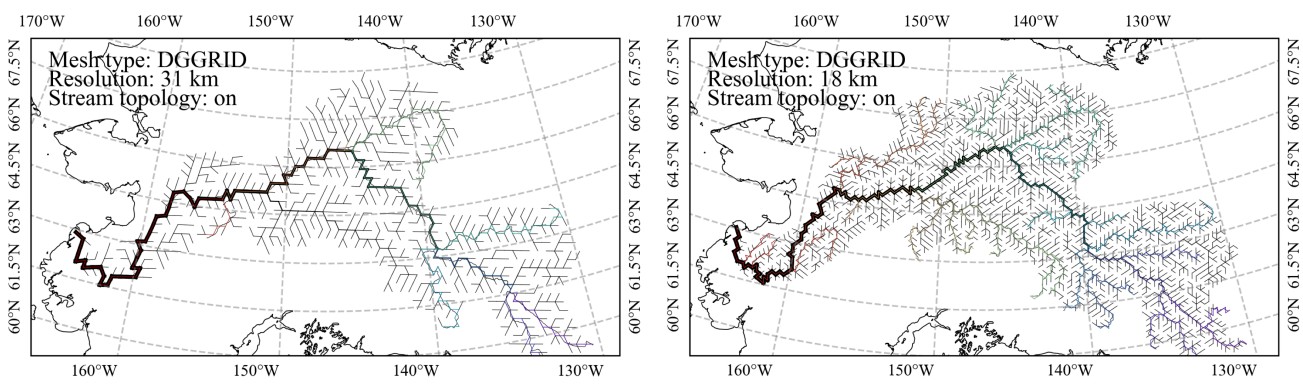

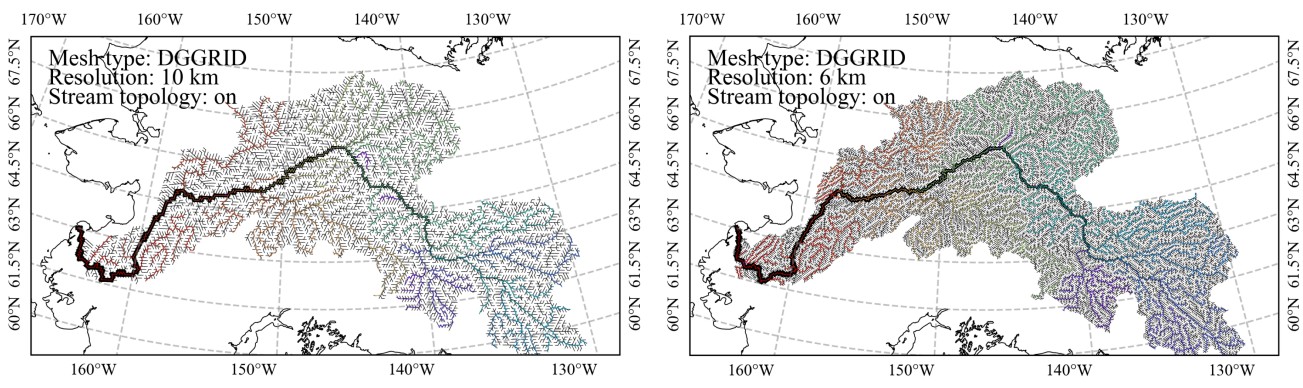

**Figure C9.** Modeled flow direction at DGGRID ISEA3H level 10 to 13 resolutions in the Yukon Basin. Black lines are cell-to-cell flow direction. Line thickness is scaled with drainage area. Colored and detailed black lines are conceptual and simplified HydroSHEDS river networks.



Drainage area

**Figure C10.** Modeled drainage area at DGGRID ISEA3H level 10 to 13 resolutions in the Yukon Basin (units: m$^2$).





**Figure C11.** Modeled travel distance to the basin outlet at DGGRID ISEA3H level 10 to 13 resolutions in the Yukon Basin (unit: m).



## Appendix D: Data validation

### D1 Tributaries along the Amazon River

| Tributary | Longitude (°) | Latitude (°) |
|---|---|---|
| Amapa Para | -51.20655 | -0.03696 |
| Ilha Urucuricaia | -52.23621 | -1.54800 |
| Santarem | -54.76348 | -2.39293 |
| Linha Rio Madeira | -58.77602 | -3.39210 |
| Manaus | -60.02855 | -3.14578 |
| Leticia | -70.00767 | -4.37262 |

**Table D1.** List of tributary outlets along the Amazon River used for drainage area and travel distance validations.

### D2 Tributaries along the Yukon River

| Tributary | Longitude (°) | Latitude (°) |
|---|---|---|
| Tanana River | -151.85660 | 65.13456 |
| Porcupine River | -141.69291 | 67.18116 |
| Koyukuk River | -157.55722 | 64.92744 |
| Stewart River | -139.39908 | 63.29666 |
| Pelly River | -137.35090 | 62.80660 |

**Table D2.** List of tributary outlets along the Yukon River used for drainage area and travel distance validations.

*Author contributions.*

Chang Liao prepared the input data and conducted the HexWatershed simulations and analysis. Darren Engwirda developed and tested the REACH library using the HydroSHEDS river network datasets. All the co-authors contributed to the writing and analysis.

*Competing interests.*

The contact author has declared that none of the authors has any competing interests.



*Acknowledgements.* This research was funded as part of the multi-program, collaborative Integrated Coastal Modeling (ICoM) project and the Interdisciplinary Research for Arctic Coastal Environments (InteRFACE) project through the Department of Energy, Office of Science, Biological and Environmental Research program, Earth and Environment Systems Sciences Division, Earth System Model Development (ESMD) program area. This work was also supported by the U.S. Department of Energy Office of Biological and Environmental Research as part of the Terrestrial Ecosystem Systems program through the Next Generation Ecosystem Experiment (NGEE) Tropics project. A portion of this research was performed using PNNL Research Computing at Pacific Northwest National Laboratory. PNNL is operated for DOE by Battelle Memorial Institute under contract DE-AC05-76RL01830. We thank Kevin Sahr for his support on the DGGRID software.





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



| Parameter name | Usage | Value | Note |
|---|---|---|---|
| dggrid_operation | Mesh generation purpose | GENERATE_GRID | |
| dggs_type | DGGs mesh type | ISEA3H | |
| dggs_res_spec | Resolution level | 10, 11, 12, 13, 14 | Level 14 is used for validation |
| clip_region_files | The clip region from the globe | The Amazon Basin boundary file | |
| update_frequency | | 10000000 | |
| cell_output_type | Output format | GDAL_COLLECTION | |
| cell_output_file_name | Output file name | dggrid | |
| densification | Point on cell edge | 0 | |
| max_cells_per_output_file | File size control | 0 | |
| neighbor_output_type | File format for neighbor information | GDAL_COLLECTION | Neighbor information stored in the mesh |

**Table A1.** List of the DGGRID mesh generation parameters used in this study.