# Peer review of "Discrete Global Grid System-based Flow Routing Datasets in the Amazon and Yukon Basins"

_Earth System Science Data, 2023_

## Author Response (AR1)

**Review #1:**

The authors describe the creation of DGGS-based hydrography data. While aiming at DGGS, the authors only mention ISEA3H without other meaningful information (index type, seqnum, z3, or other, comparing with other hexagonal DGGS types like ISEA4H or ISEA7H). Thus, DGGS integration for an advanced user would still be potentially tricky.

Reply: To address this, we provided a browser-based workshop so user can generate their own dataset using other meshes and resolutions: https://github.com/changliao1025/hexwatershed\_tutorial

These datasets are provided as (1) a protocol for hydrological models to adopt the DGGS data structure. We don't intend to generate all the meshes and resolution flow routing datasets currently due to computational demand; and (2) a final product for the Amazon and Yukon River basins to evaluate the performance of hydrologic models on the DGGS meshes.

The chosen resolutions are fairly coarse, the authors don't discuss MERIT Hydro or Hydrography90m https://hydrography.org/hydrography90m/

Reply: We targeted spatial resolutions between 1 km and 100 km for this dataset to address common spatial resolution requirements of moderate- to large-scale hydrologic models, using HydroSheds as our input dataset. While MERIT Hydro and HydroSheds both use multiple elevation datasets to achieve global coverage, they are based on the same underlying 90 m elevation data from the Shuttle Radar Topography Mission (SRTM).

It is not the goal of this study to produce another global 90 m hydrography dataset, which would be redundant with the input HydroSheds product (and MERIT Hydro and Hydrography 90m). Rather, the goal is to produce flow networks for hydrologic modeling using DGGS meshes and publicly available elevation data.

We also mentioned these two datasets as the examples of high-resolution raster-based datasets: This method often requires high-quality digital elevation model (DEM) rasters such as METIR hydro (Yamazaki et al., 2019; Amatulli et al., 2022).

Only two river basins are used, overall this could be considered a prototype or proof-of-concept, but to be a meaningful dataset there should be many more basins, and higher resolutions. The choice of data formats is awkward, GeoJSON is very inefficient, and (Geo)Parquet is very new. Better to go with e.g. GeoPackage and even some tiling or partitioning (which would go well with DGGS cells).

Reply: We agree with the reviewer that this dataset is a prototype or proof-of-concept. We hope to provide a web service in the future so that users can generate such kinds of datasets for any domain with their choice of mesh and resolution. This is still under development as it involves lots of planning in terms of input data quality and automation. Meanwhile, we provided the workshop materials that the users can customize to meet their needs.

We provide GeoJSON format for small watersheds and Geoparquet for large watersheds with high resolutions. Thanks for the suggestion on GeoPackage, we have added this format in the datasets.

Is this about a dataset or the software? the datasets do not have the quality or meaningfulness to be deposited in ESSD. The methodology is mostly well described, the initial integration of the DEM data into the hexagonal grid is not explained (I read zonal stats later somewhere?).

Reply: This study includes both datasets and the methods/software to produce them.

There are many software/models that can be used to generate DGGS-based datasets. However, none of them can generate the flow routing datasets, including the flow direction and flow accumulation. To the authors' knowledge, our workflow (with software/models) is the only resource capable of producing this type of dataset. Therefore, we provide detailed information for both models and datasets.

These datasets should be viewed as both a prototype and a final product. so that (1) the hydrologic modelers can use this unique dataset to extend existing hydrologic models so that they can support DGGS; (2) if a hydrologic model supports DGGS, then this dataset can be used to evaluate the performance of DGGS-based river routing in the Amazon and Yukon Basins.

We conducted data quality evaluation using elevation, flow accumulation, and direction. Future evaluation can be carried out to determine if any hydrologic models can run hydrologic simulations using these datasets.

Yes, the DEM integration uses the zonal mean (clipped by each DGG mesh cell). We also include a reference to this method. See "Assign elevation to the mesh cells based on raster DEM and each mesh cell boundary. A zonal mean resampling method is used by default;"

- unconventional abbrev DGGs, is it meant as DGG Discrete Global Grid's (plural) or should it be DGGS Discrete Global Grid System, with plural DGGS's?

Reply: In our manuscript, we use DGGS to represent Discrete Global Grid Systems. We have checked the whole manuscript to ensure consistency.

- citation irregular format Matthew B. J. Purss et al., 2016 ll14, (Kevin Sahr, 2019), ll 16

**Reply: Thank you for pointing out this issue. We have updated the citation.**

- unclear 1133 "..., which does not support vector-based datasets"

Reply: We revised to "This is because existing DGGS-based hydrology datasets are often derived by resampling from existing raster-based datasets, which does not support vector-based datasets such as flow direction." We mainly refer to the fact that the existing resampling methods often require the input datasets to be in raster formats (such as DEM). Therefore, they cannot be applied to common vector file formats, especially for flow direction, which is polyline-based.

- 1135 "within a DGGs-based framework"

Reply: We have revised this sentence.

- abbrev PCS ever used again?

Reply: We checked the manuscript; PCS was not used elsewhere, so we removed it from this sentence.

- 1137 "improved numerical performance for surface and subsurface hydrologic models (Liao et al., 2020)" how?

Reply: Thank you for raising this concern. We revised this sentence to "potential numerical performance improvement for coupled surface and subsurface hydrologic models"

We added additional citations to support this statement. One of the arguments is

Island effect and its impact on surface-subsurface hydrologic model

Illustration of the island effect and its impact on coupled surface-subsurface hydrologic models. The red cube/cell is an island in the surface hydrology model and is connected through a diagonal path (red arrow). The dashed cube underneath the red cube cannot be modeled in a Cartesian grid-based subsurface hydrologic model because only the top face is connected with the red cube. The green cube is often not considered a ``neighbor" of the dashed cube in the subsurface hydrologic models,

This idea that diagonal neighbor (D8) is only considered in surface hydrology but not subsurface hydrology will introduce uncertainty.

- 1139, "3) more flexibility in spatial resolution due to their hierarchical data structure" how does that hold true when compared against rasters?

Reply: Currently, most existing hydrologic models use a "raster" structure to define the spatial resolution. At the regional scale, the resolution is rather arbitrary, so it can be any resolution.

However, existing methods often constrain them at the large to global scale. For example, global-scale hydrology models use 1 degree,  $\frac{1}{2}$  degree,  $\frac{1}{4}$  degree spatial resolutions. It is often not straightforward to generate a mesh around  $\frac{1}{7}$  degree.

In contrast, with DGGS, users can generate most of the desired resolution using different combinations of DGGS and resolution levels.

- "ISEA3H" better than ISEA4H or ISEA7H? The reasoning here is only for hexagonal in general

**Reply: We changed it from ISEA3H to ISEA. They are similar in terms of the hexagonal grid structure. In our study, we only used ISAE3H as an example.**

- "ISEA streamlines calculations of conserved quantities" ... using ISEA cells, yes, ISEA itself does not streamline. Please avoid too sensationalist wording, overall the concept is sound and stands by itself.

Reply: We revised it to "As an equal-area icosahedral DGGS projection, ISEA eliminates the need for equal-area projection."

- ll44: "This study breaks new ground by developing new flow routing datasets using the ISEA3H DGGs and our newly developed mesh-independent flow direction model." really strange wording, the second part does not fit well. make separate sentence?

Reply: We revised it to "This study advances the field by developing new flow routing datasets using the ISEA3H DGGS and our innovative mesh-independent flow direction model."

- 1170: "Yukon Basin at 15-arc-second (~ 500 m) ", 15 arc seconds is slightly less than 500m at the equator, but almost half at e.g. 60 degrees, this might be something to consider when deciding for an ISEA3H resolution, please review with your methodology and elaborate.

Reply: Yes, the 15-arc-second ( $\sim$  500 m) refers to the resolution at the equator. We revised this to clarify.

- 1183: REACH library (Engwirda, 2023), does simplification use which priority, strahler or other channel order systems?

Reply: in this work we target simplification with the geometry of computational grids in mind, and therefore don't employ standard topological methods (e.g. Strahler) directly. We instead develop a greedy simplification approach that prioritizes the preservation of large-catchment streams, while filtering out short and/or geometrically too-close tributary segments that are not desirable wrt. computational grid generation:

REACH employs a greedy network simplification algorithm in which the maximal set of river reaches is processed in priority order of increasing upstream catchment area. Reaches are removed incrementally if they meet the following criteria: 1) their length is

shorter than a user-defined tolerance, or 2) they are geometrically closer to another, higher priority reach segment than a user-defined tolerance. Upon removal of a given river reach, the downstream network is simplified — merging any newly contiguous segments into 'super-reaches' and updating their associated priorities.

- Figure 2, why these resolutions shown? Not explained, ll92 "user-defined tolerance" ... at least example to reduce the feeling of arbitrariness? Why can there be isolated river segments if you redo routing and use the catchment boundary... somewhat inconsistent feeling here

Reply: There was an ordering issue in our previous manuscript. The resolutions shown in Figure 2 are from Table 1. We revised the manuscript to present Table 1 before Figure 2.

We added a sentence to explain why these resolutions are chosen: "These resolutions are chosen to accommodate major existing large-scale hydrologic and Earth System Models."

We explained the "user-defined tolerance" in the following paragraph "In practice, the user-defined tolerance is often set as the mesh cell's spatial resolution, which can vary in space as well." We added "The user-defined tolerance is a measure of how much detail the model and mesh should preserve because a mesh cell conceptually can only represent one main channel unless it is a river confluence."

For the isolated river segment issue, they are preserved in Figure 2 because after the REACH library operation at the global scale, a clipping operation was conducted for each basin, and some segments at the boundary were broken into pieces.

The whole clipped river segments are input into the HexWatershed model, which automatically removes them because they are not connected to the main river system.

However, we also think the REACH library may have an issue near the boundary, which we will improve in future development.

- ll96: "DGGRID is an open-source library developed by Kevin Sahr in 2003" the development started maybe in 2003, but it is still actively developed. Please rephrase.

Reply: DGGRID is an actively maintained, open-source library, initially developed by Kevin Sahr in 2003.

- check (Sahr, 2015), suggest additional Sahr

Reply: thank you for the suggestion, we added additional citations.

- version 7.0? not even 7.6, 7.6 or 7.8 from the last years?

Reply: We re-ran our model using the newer version of the DGGRID model. And we revised the manuscript as "The DGGRID version 8.3 was used in our study to generate the ISEA Aperture 3 Hexagonal (ISEA3H) meshes with the default orientation."

- table 1: Showing internode spacing (distance between centroids?) and a square root of area (for a close-to-circle shape) is probably not meaningful (they are almost the same for the purpose here). You could rather keep the actual cell area in km2 or ha, and clarify that internode spacing could also be interpreted as analog to "pixel" size of conventional raster, and that it closely corresponds to a diameter of a circle of the same area, so the reader can understand the spatial resolution of the grid cells.

Reply: We changed the root of area to area.

We decided to keep the internode spacing to ensure consistency between our method and the DGGRID definition. The resolution in length is also used for visualization.

- p6/7: DGGRID Application Programming Interface (API) (Liao, 2022a)? ... "implemented several APIs to set up a DGGRID model run"

Reply: Once the DGGRID model is built from its C/C++ source code, it can be called through several Application programming interfaces (APIs), which have been implemented within the HexWatershed model.

- 11139: " topological relationship-based reconstruction method" is that explained somewhere under the same term?

Reply: HexWatershed includes two components, both of which use the topological relationship-based method, but they refer to different algorithms.

In the topological relationship-based river network representation method it mainly refers to the topological relationship-based reconstruction algorithm.

In the depression removal component, it mainly means the topological relationship-based stream burning method.

- 11154: GCS and GeoJSON, which GCS, probably WGS84, EPSG:4326? please name it explicitly.

Reply: We add the "EPSG:4326" into the sentence.

**"All the spatial datasets are provided using the GeoJSON file format with the GCS WGS84 EPSG:4326 spatial reference"**

- in section 3 all subsections start with the same "The variable\_polygon.geojson is a polygon-based GeoJSON data file", I think that can be introduced once, and then does not need to be repeated for each variable in that file. On another note, using GeoJSON for these larger types of data feels fairly ineffective and wastes a lot of storage space, why not use something robust like GeoPackage or FlatGeobuf? Everything beyond 5000-10000 cells/geometries people are not loading into a browser, so why GeoJSON?

**Reply: Thanks for the suggestion. Currently, the workflow outputs GeoJSON files by default, which are converted to GeoPackage and GeoParquet for performance considerations.**

- 11177: "special attention was paid to the consistency across different spatial resolutions." How? Consistency is not really elaborated on in the following sections.

Reply: By "consistency across different spatial resolutions", we generally mean that the workflow should perform reasonably well regardless of spatial resolution. This is the default design of "mesh independent". Because mesh, in principle, can have any spatial resolution, the workflow must be robust so it can accept most resolutions without special treatments or manual corrections.

For example, the traditional DEM-based watershed delineation is not consistent because if the DEM is very coarse, the resulting flow direction map can be significantly different and even incorrect.

We removed this sentence to avoid confusion.

- 11185: To which "zonal mean resampling procedure" are you referring here, please be specific (refer to methods sections), and which biases? please elaborate

Reply: We added a citation to the zonal mean resampling method. This method was introduced earlier in the method section:

"Assign elevation to the mesh cells based on raster DEM and each mesh cell boundary. A zonal mean resampling method is used by default;"

- Figure 13 and 14: When you name "modeled surface elevation" you refer to the elevation in the hex grid cells, which have been post-processed (depression filling etc) after their initial "ingestion" (from which source again?)

Reply: Yes, the original elevation is a raster. After the zonal mean resampling, each mesh cell has one elevation value. After the stream burning/depression filling, the elevation value is updated.

- ll223 " use high-resolution meshes (such as the ISEA3H resolution level 13) " ? the 4 resolutions in the paper are (29.42 km, 16.99 km, 9.81 km, and 5.66 km) how is that "high-resolution" when Hydrosheds data is available at 500m / 1km scales?

Reply: This statement is in the context of large-scale hydrological and an Earth system models. We revised it to clarify.

"To address this, it is recommended to use high-resolution meshes (such as the ISEA3H resolution level 13) to mitigate its impact on model performance for large-scale hydrologic and Earth system models."

- ll241 " Topographic Wetness Index (TWI)" ... how can a user derive these indicators from the hex mesh dataset?

Reply: By definition, the calculation of TWI requires an upslope drainage area and local slope. Both of them are generated by the model with some assumptions. Therefore, model users may derive TWI if needed. However, this TWI dataset will also be DGGS-based.

- Table A2 why does flow direction data type not applicable? Units would also be good, km2 or m, slope in degrees or percent ... etc.

Reply: unlike raster-type flow direction that uses 1 2 4 8, et al., to indicate flow direction, unstructured mesh-based flow direction does not use numbers to indicate flow direction. Instead, our datasets used the upslope and downslope mech cell global IDs to represent direction.

Review #2:

The paper develops new flow routing datasets using Icosahedral Snyder Equal Area (ISEA) DGGs and a novel mesh-independent flow direction model. Technical validation evaluates the proposed datasets from the surface elevation, flow direction, and drainage area. The paper demonstrates the potential of DGGS in the field of hydrologic models.

Comments:

1. The authors conducted a visual examination of the modeled flow directions using the simplified HydroSHEDS river networks in section 4.2. The drainage area was used to validate these flow directions. Actually, characteristic lines of terrain can also be used for

validation, such as areas of differences between the fitting valley lines and true lines. If feasible, it is recommended to include more quantifiable

Reply: Thank you for the comment. We have to clarify that flow direction is slightly different from river networks. This is because not all mesh cells have a well-established river channel inside due to its dependency on the mesh resolution.

However, our study produces both river networks and flow direction in separated steps. The conceptual river networks are produced by the PyFlowline component. Because hydrologic models usually use the flow direction data which has river networks embedded already, we only present flow direction earlier.

However, in this revision, we added the river network validation using the same approach we used in our earlier studies.

2. The authors verified the proposed dataset using surface elevation, flow direction, drainage area and travel distance. However, they did not mention the time cost of constructing the dataset. A good algorithm should balance both computational efficiency and accuracy. As the subdivision level increases, the number of cells in DGGS grows exponentially, significantly impacting the computational efficiency. It is recommended that the authors include a discussion on computational efficiency.

Reply: Thank you for this comment. Indeed, computational performance is a key aspect of the modeling component, and we didn't provide many details. In this revision, we added more discussion about this.

3. There is a limitation on watershed extraction in flat areas. The authors did not discuss this issue. It is recommended to include relevant discussion in the Technical Validation

Reply: Thank you for the comment. This is a known issue in most existing watershed delineation tools/models. The stream-burning technique developed in HexWatershed is our current approach to mitigate this issue. When a river network is burned in the workflow, the model will guarantee that the surrounding area of the river network will be included, even if it is in flat areas.

Minor comments:

Unable to directly link to relevant figures by the contents in the manuscript.

Reply: Thank you for the comment. We have checked the manuscript again to ensure that all the figures are referred to appropriately.